# Immune alterations in subacute sclerosing panencephalitis reflect an incompetent response to eliminate the measles virus

Sibel P. Yentür[1], Veysi Demirbilek[2], Candan Gurses[3¤a], Safa Baris[4], Umit Kuru[5], Semih Ayta[6], Zuhal Yapici[3], Suzan Adin-Cinar[7], Serap Uysal[8], Gulden Celik Yilmaz[9¤b], Emel Onal[10], Ozlem Cokar[6], Güher Saruhan-Direskeneli[1] *

1 Department of Physiology, Istanbul Medical Faculty, Istanbul University, Istanbul, Turkey, 2 Department of Neurology, Cerrahpasa Medical Faculty, Istanbul University Cerrahpasa, Istanbul, Turkey, 3 Department of Neurology, Istanbul Medical Faculty, Istanbul University, Istanbul, Turkey, 4 Department of Pediatrics, Marmara Medical Faculty, Marmara University, Istanbul, Turkey, 5 Department of Pediatrics, Bayrampasa State Hospital, Istanbul, Turkey, 6 Department of Neurology, Haseki State Hospital, Istanbul, Turkey, 7 Department of Immunology, Aziz Sancar Institute of Experimental Medicine, Istanbul University, Istanbul, Turkey, 8 Department of Pediatrics, Cerrahpasa Medical Faculty, Istanbul University Cerrahpasa, Istanbul, Turkey, 9 Department of Microbiology, Istanbul Medical Faculty, Istanbul University, Istanbul, Turkey, 10 Department of Public Health, Istanbul Medical Faculty, Istanbul University, Istanbul, Turkey

¤a Current address: Koc University Medical Faculty, Istanbul, Turkey
¤b Current address: Bahcesehir University, Faculty of Medicine, Istanbul, Turkey
* gsaruhan@istanbul.edu.tr

**Data Availability Statement:** All relevant data are within the manuscript.

## Abstract

In subacute sclerosing panencephalitis (SSPE) the persistence of measles virus (MeV) may be related to the altered immune response. In this study, cytokine responses of lymphocytes and monocytes were evaluated in SSPE compared to controls with non-inflammatory (NICON) and inflammatory (ICON) diseases. Patients with SSPE (n = 120), 78 patients with ICON and 63 patients with NICON were included in this study. Phenotypes of peripheral blood mononuclear cells (PBMC) have been analyzed by flow cytometry. CD3 and CD28, and *S. aureus* Cowan strain I (SAC) stimulated and unstimulated cells were cultured and IL-2, IL-10, IFN-γ, IL-12p40, IL-12p70 and IL-23 were detected in supernatants by ELISA. MeV peptides were used for MeV-specific stimulation and IFN-γ secretion of PBMC was measured by ELISPOT. Spontaneous and stimulated secretions of IL-10 were lower in SSPE compared to both control groups. T cell stimulation induced lower IFN-γ production than ICON group, but higher IL-2 than NICON group in SSPE. Stimulated PBMC produced lower IL-12p70 in SSPE and had decreased CD46 on the cell surface, suggesting the interaction with the virus. IFN-γ responses against MeV peptides were not prominent and similar to NICON patients. The immune response did not reveal an inflammatory activity to eliminate the virus in SSPE patients. Even IL-10 production was diminished implicating that the response is self-limited in controlling the disease.

**Funding:** The study is supported by Istanbul University Research Fund (BAP: ACIP3363-ACIP-22421-T291).

**Competing interests:** The authors have declared that no competing interests exist.

## Introduction

Subacute sclerosing panencephalitis (SSPE) is a progressive disease of the central nervous system (CNS) affecting mainly children and early adolescents. It is a rare and late complication of measles virus (MeV) infection with fatal outcome. Typically SSPE patients have a history of primary measles infection at an unusually young age followed by a latent period of 6 to 8 years. The incidence of SSPE worldwide is estimated 1 per million [1]. In a recent epidemiological study of Istanbul, the incidence was found as 2 per million with a girl dominance [2].

Both alterations in the host immune system and changes in the MeV have been the subject of investigations on the pathogenesis of SSPE. In early studies, antibodies against nucleocapsid and matrix proteins of the virus were detected in the serum and cerebrospinal fluid (CSF) of patients. Proliferative T cell responses were found comparable to healthy individuals [3]. However, virus-specific cytotoxic activity was impaired while NK cell cytotoxicity was preserved [4]. Suppression of Th1 cytokine production was reported by demonstrating defective IFN-γ response of peripheral blood mononuclear cells (PBMC) to MeV in SSPE patients with severe disease progression [5]. Elevated IL-12p70+p40 and CXCL10 levels in CSF without an accompanying IFN–γ increase compared with other inflammatory and non-inflammatory disease controls were reported [6]. Higher serum IL-2 concentrations and suppressed Th2 cytokines (IL-4, IL-6 and IL-10) were also demonstrated [7]. Lower IL-12 secretion in response to MeV vaccine and PPD, and IFN-γ and IL-10 productions to PPD have also provided evidence for alterations in immune response regulation in a previous study [8]. More recently, activated IL-12/IFN-γ and the IL-23/IL-17/IL-22 pathways were reported in SSPE patients [9].

The virus is anticipated to induce alterations in the immune response of infected cells to MeV. As a costimulatory molecule of the immune system, signaling lymphocyte activating molecule (SLAMF1/SLAM/CD150) is a common receptor for MeV and expressed mainly on activated lymphocytes [10–12]. In MeV infection, SLAM expression is reduced on host cells [13]. On the other hand, CD46 has been shown to act as a receptor mainly for a limited number of MeV strains [14]. The binding of CD46 by MeV has reduced IL-12 production of monocytes thereby regulating the immune response against the virus [15].

In the present study, the immune response to MeV in SSPE patients has been investigated by cytokine secretions of PBMC in response to non-specific and specific stimulations. Possible regulatory changes on the immune response have been evaluated mainly by cytokine measurements.

## Materials and methods

### Patients and controls

The study was approved by the institutional ethics committee of Istanbul University Istanbul Medical Faculty and written informed consent was taken from the patients and from the parents of the children according to the Declaration of Helsinki. Patients with SSPE (n = 120) were enrolled to the study between the years 2003–2017. Seventy-five of them were male and 45 were female (Median age: 9 years (1–34 years)). All patients fulfilling the criteria for the diagnosis of SSPE are included in this prospective study: Typical clinical presentations (myoclonus, head drops, hemiplegia, deafness, severe mental and behavioral changes, dementia, visual and speech involvement), EEG findings, measles antibody titers in the CSF [1, 16, 17]. Additionally, all 108 patients tested had oligoclonal IgG bands; in 17.6% of patients presented pattern 2, while in 82.4% patients patterns 3 and 4 with concomitant IgG bands in the serum were detected. Only the patients whose parents did not consent are excluded. The patients were referred to the specialized neurology centers of the referral hospitals from Marmara

Region and also from Black Sea, East and Southeast Anatolia regions of Turkey. According to the limited information available from the patients, only 56.7% of patients had a natural history of measles disease occurring between 2 months and 9 years of age and 46.7% of the patients had a known history of measles vaccination (Table 1).

Blood from 141 totally unrelated donors were used as controls: 78 of them had inflammatory diseases (ICON) such as multiple sclerosis, asthma, bronchitis, Miller-Fisher syndrome, type I diabetes, tonsillitis, upper respiratory tract infection and viral infection (37 female and 41 male, median age: 8 years (1–24 years)). As another control group, 63 patients (20 female and 43 male, median age: 11 years (1.5–38 years)) with non-inflammatory diseases (NICON) like afebrile convulsion, anemia, headache, leg pain, fatigue, joint pain, epilepsy, X-linked adenoleucodystrophy, nesioblastosis, vomiting and urticeria were included in this study (Table 1). SSPE patients and control donors were not on immunomodulatory treatment.

Due to the scarcity of the blood obtained from the donors, not all measurements have been performed in all donors. Measles antibodies (IgG) were detected with ELISA kit (2326000, Trinity Biotech, Ireland and ESR102G, Serion, Institut Virion, Germany) in all CSF samples.

## Phenotypic staining and reagents

PBMC of donors were isolated from EDTA anti-coagulated blood samples by ficoll density gradient centrifugation. Cells were surface stained with fluorochrome-conjugated mouse anti-human CD3-FITC and -APC (IgG1, A07746, Beckman Coulter, France and C7225, Dako Cytomation, Denmark), CD8-APC (IgG1, C7227, Dako Cytomation, Denmark), CD4-APC and -FITC (IgG1, C7226 and F0766 Dako Cytomation, Denmark), CD19-PC5 (IgG1, A07771, Beckman Coulter, France), CD14-FITC and -PC5 (IgG2a, F0844, Dako Cytomation, Denmark and A07765, Beckman Coulter, France), CD45-FITC/CD14-PE (IgG1/IgG2a, Catalog Nr: 873.032.050, Diaclone, France), CD46-PE (IgG2a, 197–050, Ancell, USA), CD150-PE (IgG1, 12–1509, eBioscience), PD-1-PE (IgG1, 557946, BD Pharmingen, USA) and isotype control antibodies (BD Biosciences Pharmingen, USA and X0950, X0933, X0968, Dako Cytomation, Denmark and A07798, Beckman Coulter, France). Phenotypes have been analyzed by flow cytometry (FACSCalibur, Becton-Dickinson) and compared between groups in respective cell gates.

## Cell stimulation and measurements of proliferation and cytokines

To stimulate the T cells, flat-bottomed 96-well plates (TPP, Switzerland) were coated overnight at 4˚C with anti-CD3 (10 μg/ml, 854.010.000, Diaclone, France) and anti-CD28 (5 μg/ml, 177–020, Ancell, USA) or with IgG1 isotypic control (5 μg/ml, 857.070.000, Diaclone, France) antibodies. PBMC were seeded as 200.000 cells/well in triplicates. In another set of donors, *S. aureus* Cowan strain I (SAC, Pansorbin, 507858, Calbiochem, USA) was added to a final dilution of 1:10000. After 72 hours of incubation at 37˚C in 5% $CO_2$ with culture medium containing RPMI-1640 (R0883, Sigma, USA), 10% FBS (10082139, Gibco, USA), 100 IU/100 μg/ml

**Table 1. Characteristics of donors.**

| Group | N (M/F) | Age (years) | Disease onset age | Measles vaccination | Measles | Age of infection (months) |
|---|---|---|---|---|---|---|
| SSPE | 120 (75/45) | 9 (1–34) | 9 (3–19) | 46.7% | 56.7% | 18 (2–108) |
| ICON | 78 (41/37) | 8 (1–24) | 6 (2–15) | 43.6% | 7.7% | |
| NICON | 63 (43/20) | 11 (1.5–38) | 6.5 (1–16) | 47.6% | 6.4% | |

SSPE: Subacute sclerosing panencephalitis patients, ICON: Controls with inflammatory diseases, NICON: Controls with non-inflammatory diseases. Ages were presented as median values, and minimum and maximum values were given in parentheses. M: male, F: female.

penicillin/streptomycin (P4333, Sigma, USA) and 2 mM L-glutamine (25030081, Gibco, USA), supernatants were collected and fresh medium was added. Proliferation was detected by [$^3$H] thymidine incorporation (0.5 µCi/well, 20 Ci/mmol, ART 178C, American Radiolabeled Chemicals, USA) after overnight incubation in culture. Supernatants were stored at -80˚C and used for the detection of IL-2, IL-10, IFN-γ (KHC0022, CHC1323, CHC1233, Biosource, USA), IL-12p40 and IL-12p70 (551116 and 559258, BD Biosciences, USA) and IL-23 (BMS2023, Bender MedSystems, Austria) by ELISA according to manufacturer's protocols.

Peptides from hemagglutinin (H, 30–38), matrix (M, 211–219), C protein (C, 84–92), two of nucleoprotein (N1, 210–218 and N2, 340–348) and the pool of these peptides (MeVp) were used to MeV specific stimulations [18]. All peptides and the MeVp were used as 10 µM (NMI Peptides, Germany). IFN-γ secretion of PBMC was detected with ELISPOT (3420-2AW-Plus, MabTech, Sweden and 874.000.005 Diaclone, France) according to manufacturer's guidelines. PHA (5 µg/ml, Sigma) was used as positive control and spot counts were presented as spots/200000 cells (CTL Europe GmbH).

## Statistical analysis

Statistical analyses were performed by non-parametric tests (Anova and Mann-Whitney U tests) for comparisons between groups using SPSS. Results were presented as median values. A p value < 0.05 was regarded as significant.

## Results

### Cytokine responses in SSPE

When we evaluated the distributions of T and B cells, and monocytes in the peripheral blood of SSPE patients with two different age-matched control groups with (ICON) or without inflammatory diseases (NICON), the proportion of CD3$^+$ T cells was slightly decreased in SSPE patients compared only to NICON (64.7% vs. 68.9%, p = 0.02), confirming our previous findings [19]. No other alterations were observed in the distribution of CD4$^+$ and CD8$^+$ T cells, CD19$^+$ B cells and CD14$^+$ monocytes in all groups (Table 2).

**Spontaneous cytokine secretion.** To analyze *in vivo* stimulated state of PBMC, we firstly measured spontaneous secretion of IL-2, IFN-γ, IL-12 and IL-10 in cell culture. Among all measured cytokines, only IL-10 levels were significantly lower in SSPE compared with both ICON and NICON groups (4.1 vs. 29.8 and 44.5 pg/ml; p = 0.014 and p = 0.002) (Fig 1).

**Table 2. Distribution of CD3$^+$, CD4$^+$ and CD8$^+$ T cells, CD19$^+$ B cells and CD14$^+$ cells among PBMC in the study groups.**

|  | CD3$^+$ cells | CD4$^+$ cells | CD8$^+$ cells | CD19$^+$ cells | CD14$^+$ cells |
|---|---|---|---|---|---|
| **SSPE** | 64.7* | 32.4 | 26.1 | 11.6 | 73.9 |
|  | (15.5–82.7) | (11.4–62.8) | (9.0–45.0) | (1.7–25.7) | (2.4–99.1) |
| **N** | *50* | *90* | *89* | *27* | *62* |
| **ICON** | 66.6 | 34.3 | 25.2 | 10.2 | 76.0 |
|  | (0.4–82.5) | (11.8–53.7) | (7.9–47.9) | (2.6–17.5) | (12.1–98.8) |
| **N** | *44* | *64* | *66* | *22* | *55* |
| **NICON** | 68.9* | 36.3 | 27.1 | 7.0 | 71.8 |
|  | (29.2–80.5) | (12.5–59.2) | (15.7–52.8) | (3.8–16.0) | (5.5–96.0) |
| **N** | *22* | *33* | *36* | *9* | *26* |

Results are presented as median values, and minimum and maximum values are given in parenthesis. SSPE: Subacute sclerosing panencephalitis patients, ICON: Controls with inflammatory diseases, NICON: Controls with non-inflammatory diseases.

*p = 0.02.

**T cell receptor mediated stimulation by CD3 and CD28.** To evaluate the response of T cells by stimulation via T cell receptor, PBMC were incubated with anti-CD3 and anti-CD28 antibodies for 72 hours and cytokine productions were compared between the groups. With stimulation, IL-10 secretion was still lower in patients than in ICON and NICON groups (5.4 vs. 40.2 and 39.5 pg/ml; p = 0.002 and p = 0.012). Similarly, T cell stimulation induced lower levels of IFN-γ production compared with ICON group only (3.8 vs. 201 pg/ml, p = 0.023). However, IL-2 secretion of T cells was higher than that of NICON group (18.7 vs. 0.0 pg/ml; p = 0.005) (Fig 2).

As a costimulatory molecule possibly effecting the cytokine production and a receptor for the MeV, we measured the expression of SLAM on CD4$^+$, CD8$^+$ T cells and B cells. SLAM on CD4$^+$ T cells was slightly higher in SSPE and ICON groups without reaching statistical significance. On CD19$^+$ B cells, SLAM was mainly increased in ICON group (25.29% p = 0.033 and p = 0.04). Relatively higher levels of SLAM in SSPE patients (14.6%) were not significantly different from NICON group (2.1%) (Fig 3, S1 Fig).

Despite the differences in cytokine productions, no significant changes between the groups were observed in proliferative responses of T cells to CD3 and CD28 stimulation (S2 Fig).

**Stimulation with SAC.** Cytokine production by PBMC was also studied by SAC stimulation to cover a broader group of cell responses [20, 21]. In SSPE patients, SAC induced lower IL-12p70 production compared with ICON and NICON groups (5.3 vs. 10.5 and 9.7 pg/ml, p = 0.043 and p = 0.013). Higher IL-12p40 secretion was detected in ICON patients compared with NICON group (283.4 vs. 101.4 pg/ml, p = 0.011). No other differences in IL-10, IL-23 and IFN-γ productions were detected between the groups in response to SAC (Fig 4).

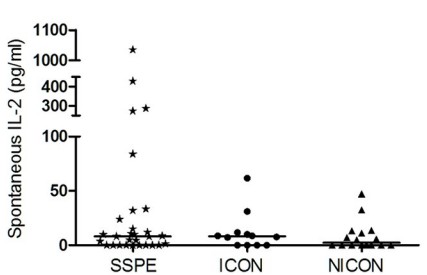
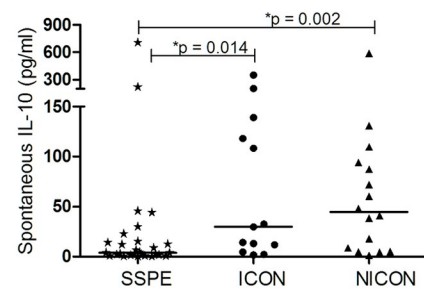
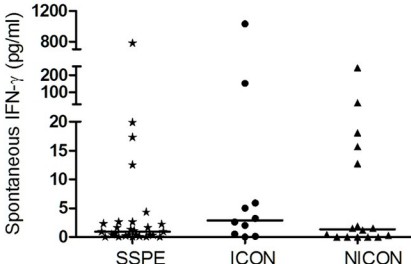
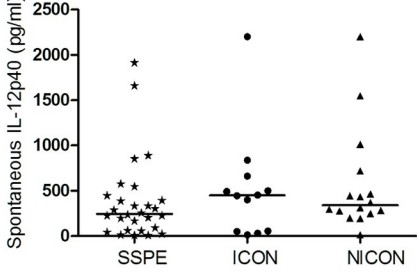
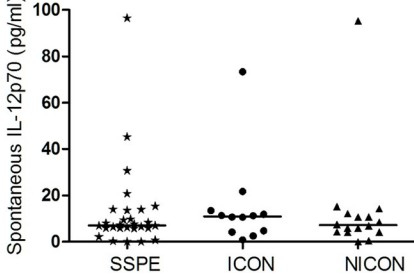

**Fig 1. Spontaneous cytokine secretion of PBMC.** Spontaneous in vitro IL-2, IL-10, IFN-γ, IL-12p40 and IL-12p70 secretion of PBMC from subacute sclerosing panencephalitis patients (SSPE, n = 29), controls with inflammatory diseases (ICON, n = 13) and with non-inflammatory diseases (NICON, n = 16) are shown. Horizontal lines depict median values.

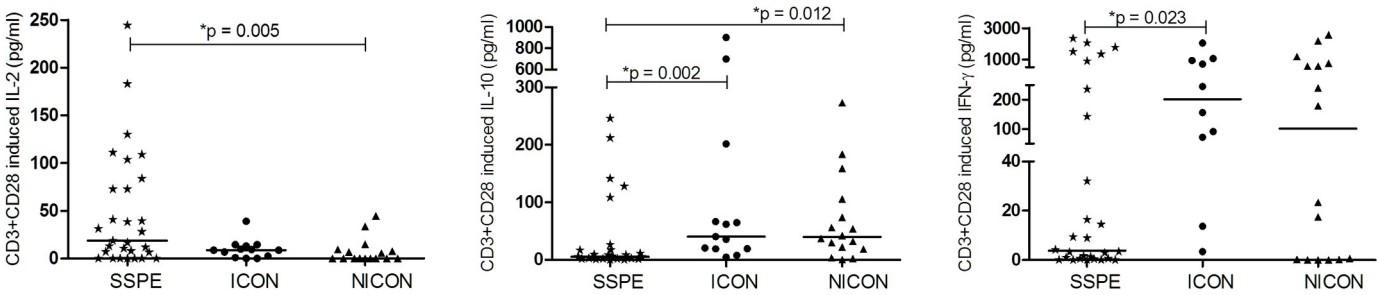

**Fig 2. CD3 and CD28 induced cytokine secretion of PBMC.** CD3 and CD28 induced productions of IL-2, IL-10, IFN-γ in subacute sclerosing panencephalitis patients (SSPE, n = 29), controls with inflammatory diseases (ICON, n = 13) and with non-inflammatory diseases (NICON, n = 16) are shown. Horizontal lines depict median values.

As the main source of IL-12 is monocytes, the reduced IL-12 production may have been induced by interaction of the virus with cellular receptors/molecules on monocytes [15]. Therefore, we screened the possible MeV receptor, CD46 on monocytes (CD14+). CD46 was lower in SSPE patients than NICON (63.0% vs. 74.4%; p = 0.033), but not than the ICON group (69.3%) (Fig 5, S3 Fig).

## Antigen-specific T cell stimulation

As the etiological agent of SSPE, the specific response against MeV has also been screened and compared with diseased controls. We have evaluated the antigen specific IFN-γ responses against immunodominant MeV peptides which were described previously [18]. SSPE patients had similar reactivity to N1, N2, M, H and C peptides of MeV and MeV peptide pool as in NICON patients. Interestingly, IFN-γ responses to all peptides and peptide pool were reduced in ICON group compared to SSPE, and NICON groups (Fig 6).

## Expression of programmed cell death-1 (PD-1) on CD8+ T lymphocytes

As a possible mechanism of viral persistence of MeV in SSPE, we have looked at the exhaustion of T cells by analyzing the expression of PD-1 on CD8+ T cells. PD-1 is a member of the CD28 superfamily that delivers negative signals upon interaction with its ligands. However, CD8+ PD-1+ cells were not different between groups in PBMC (S4 Fig).

## Discussion

The pathogenesis of the persistence and the late complication of MeV infection in the CNS have not been elucidated. In this study, the immune alterations in the SSPE patients have been

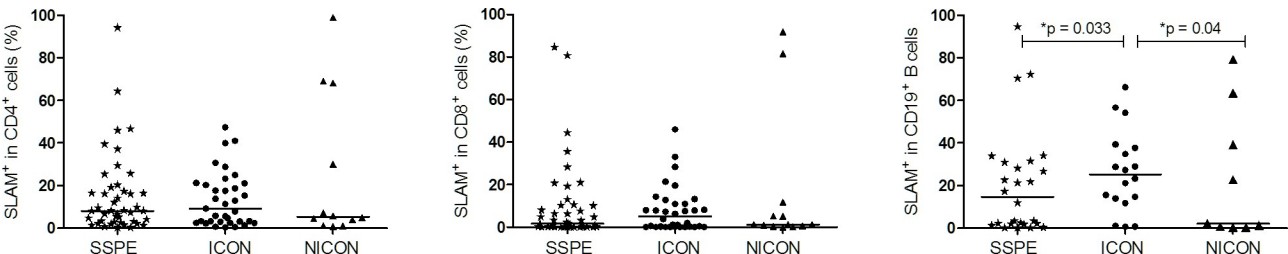

**Fig 3. Signaling Lymphocyte Activating Molecule (SLAM) expression on T and B lymphocytes.** SLAM expression on CD4+, CD8+ T cells and CD19+ B cells of subacute sclerosing panencephalitis patients (SSPE, n = 44, n = 43 and n = 26), controls with inflammatory diseases (ICON, n = 35, n = 34 and n = 18) and with non-inflammatory diseases (NICON, n = 12 n = 13 and n = 9) are shown. Horizontal lines depict median values.

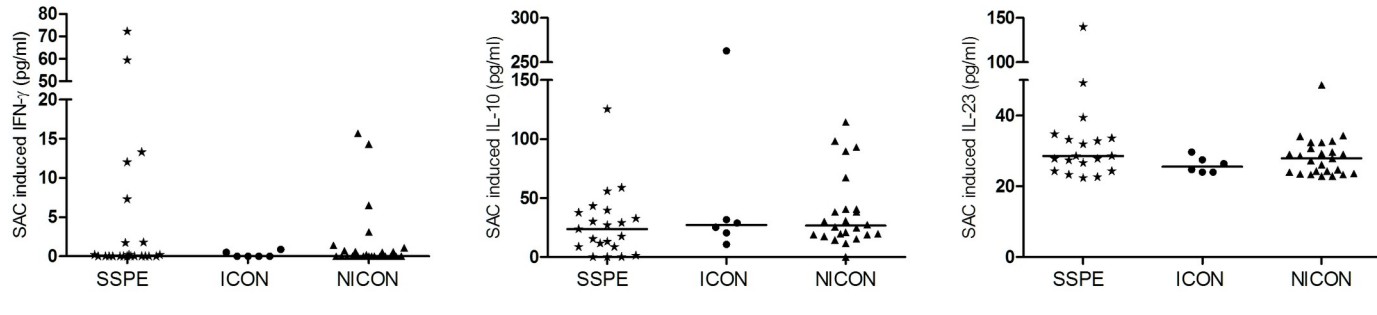

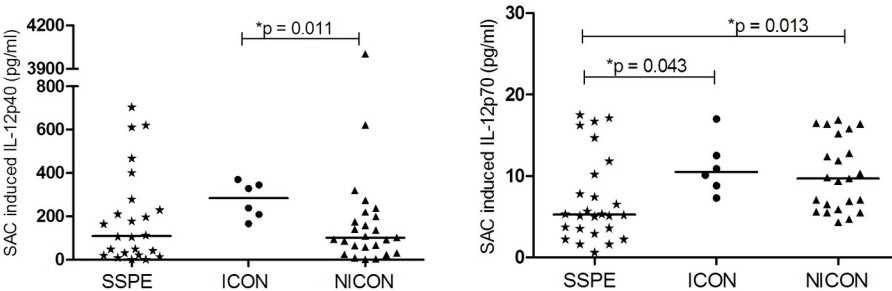

**Fig 4. SAC induced cytokine productions.** IFN-γ, IL-10, IL-23, IL-12p40 and IL-12p70 productions of SAC stimulated PBMC in subacute sclerosing panencephalitis patients (SSPE, n = 26), controls with inflammatory diseases (ICON, n = 6) and with non-inflammatory diseases (NICON, n = 25) are shown. Horizontal lines depict median values.

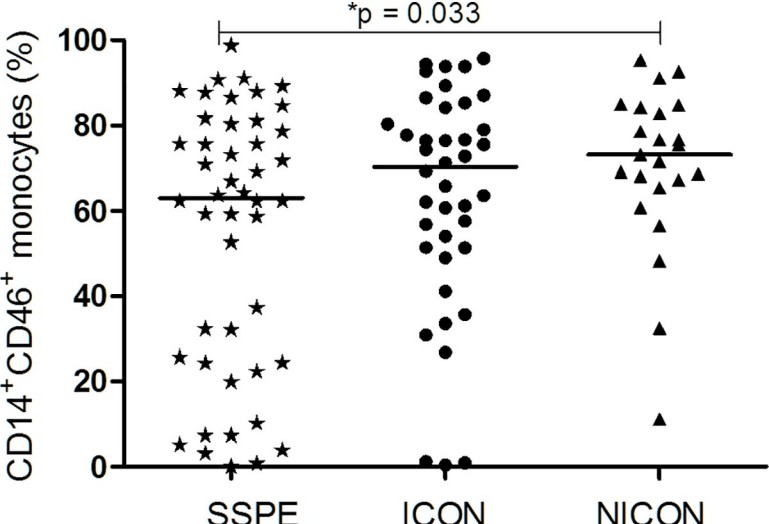

**Fig 5. CD46 expression on CD14+ monocytes.** CD46 on CD14+ cell populations was analyzed in subacute sclerosing panencephalitis patients (SSPE, n = 46), controls with inflammatory diseases (ICON, n = 40) and with non-inflammatory diseases (NICON, n = 23). Horizontal lines depict median values.

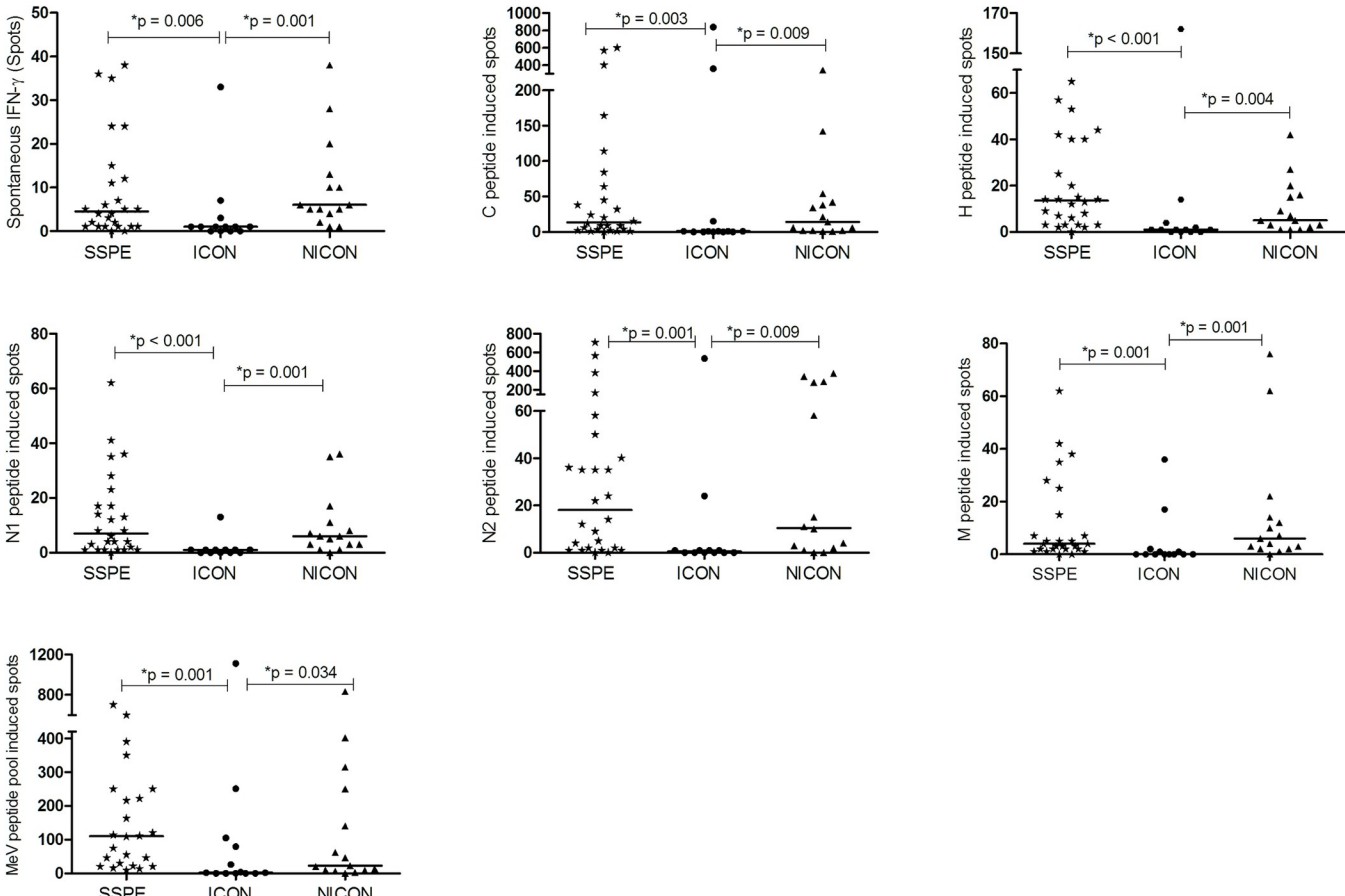

**Fig 6. MeV peptide induced IFN-γ production of PBMC.** IFN-γ secretion without and with MeV peptide (C, H, N1, N2, M and the pool) induction of PBMC in subacute sclerosing panencephalitis patients (SSPE, n = 26), controls with inflammatory diseases (ICON, n = 13) and with non-inflammatory diseases (NICON, n = 15) are shown. Horizontal lines depict median values of IFN-γ secreting cells per 200000 PMBC.

compared with two different age-matched control groups, namely patients with other inflammatory- or non-inflammatory-diseases. The immune response against MeV-specific and non-specific stimuli was evaluated in these groups. Changes were detected in cytokine secretion in SSPE compared with both control groups. Notably, cytokine production pattern of SSPE patients was relatively similar to that of NICON group. Thus, the findings of the present study emphasized the moderate immune response in the patients.

MeV infection induces profound and prolonged abnormalities in cellular immune responses in infected hosts causing an immunosuppression [22]. The mechanistic models underlying this suppression have been proposed as altered cytokine profiles, bystander lymphocyte apoptosis and lymphocyte infection and depletion [23]. The immunosuppression associated with MeV could underlay the viral persistence and the development of SSPE [4, 8]. The findings of this study did not provide evidence for an active immunosuppressive mechanism similar to acute measles infection in SSPE development.

In previous evaluations of the immune response, mainly CD4[+] T cells, B cells as well as IL-1β, IL-2, IL-6, TNF-α, lymphotoxin and IFN-γ have been detected in brain specimens from SSPE patients, which demonstrated cellular infiltrates, demyelination, gliosis and inflammatory activation [24–27]. Detection of higher IL-10 levels in both CSF [6, 28], and serum of SSPE patients [29], supported a suppressive environment in this disease. In another study, CSF

IL-4 and IL-6 concentrations were lower, whereas serum IL-2 concentration was higher in SSPE patients implicating the dominance of Th1 over Th2-type cytokines particularly at the early inflammatory response in SSPE [7]. On the other hand, preserved IL-10 production by PBMC of SSPE patients was reported, whereas defective IFN-γ secretion has been related to the worse progression of disease in response to MeV vaccine [5]. In our previous study, proliferation, as well as IFN-γ, IL-12 and IL-10 productions in response to MeV vaccine were not different from controls, although the response to purified protein derivate (PPD) was impaired in SSPE patients [8]. The decrease in regulatory T cell phenotype and the lower inhibitory NK receptors on CD8[+] T cells, and higher activating NK receptors on NK cells indicated also an active state of the immune response possibly caused by the chronic stimulation with viral antigens [19]. A recent report demonstrated increased productions of IL-12, IL-23, IL-17 and IL-22 and higher frequencies of IL-17 and IFN-γ producing cells in response to MeV peptide stimulation in SSPE pointing at both Th1 and Th17 responses [9]. In the present study, production of IL-10 was lower in SSPE compared with both control groups spontaneously and after T cell stimulation. Supporting an immune activity potential, IL-2 response of T cells was higher than in NICON group, whereas IFN-γ production was not as high as in other inflammatory diseases. The MeV peptide specific IFN-γ responses were also similar to NICON patients and not higher. This inefficient IFN-γ response in our cohort is not in accordance with previous findings, but in accordance with the lower IL-12 production. This discrepancy may be related to the selection of our patients with late stages or to the small numbers of patients in different experimental settings in these studies.

Similar to measles infection, a reduction of T cells in SSPE samples was observed in the present and previous studies [19, 30]. This observation may be related to the lymphopenia caused by MeV during the infection. MeV infected monocytes induce apoptosis in uninfected T cells which also may contribute to the pathogenesis of MeV-induced immunosuppression [31].

IL-10 is considered a prototypical anti-inflammatory cytokine, which significantly contributes to the maintenance and reestablishment of immune homeostasis. However, with its pleiotropic roles, it can also promote immune responses by supporting B cell and CD8[+] T cell activation. In infectious diseases, it is considered as a master regulator of immunity, as it can mainly help to ameliorate the excessive Th1, Th2 and CD8[+] T cell responses in infections [32]. Functionally, reduced production of IL-10 with IFN-γ in response to general T cell stimulation in SSPE may highlight the absence of a highly pro-inflammatory state of the disease. On the other hand, polymorphic features of IL-10 gene may have an effect on susceptibility to SSPE and the decreased production may be related to the low-producer alleles of IL-10 [21], which has not been investigated in SSPE.

The MeV cell entry receptor, SLAM is classified as a costimulatory molecule that favors lymphocyte proliferation, Ig synthesis, and secretion of IFN-γ [33–35]. Therefore, the interaction of MeV with SLAM could have effects on those immune cells leading to virus-mediated activation. In a previous study, SLAM expression was also demonstrated to be increased in lymphocytes, monocytes and brain tissues of two SSPE patients [36]. However, in the present study SLAM on T or B cells was not increased significantly in SSPE.

On the other hand, MeV has been shown specifically to ablate IL-12 production by monocyte/macrophages *in vitro* through binding to CD46 [15]. Reduced IL-12 production of monocytic cells was also accompanied by the down-regulation of CD46 from the surface of cells infected with vaccine strains of MeV *in vitro* [37, 38]. Lower expression of CD46 in lesions of SSPE brains suggested also an interaction of CD46 with a SSPE-specific MeV strain [14]. In SSPE patients, the lower frequencies of CD46[+] monocytes compared with NICON group in this study also implicated a related interaction of the virus with these cells. The presence of

viral RNA has been documented in monocytes during acute measles infection [39]. Although not confirmed in SSPE patients, persistence of viral RNA has been demonstrated in PBMC and lymph nodes of the monkeys experimentally infected with MeV after months [40]. Possibly a low number of MeV present in the body and inducing the stable antibody response may use CD46 to enter the monocytes and internalize these molecules persistently. Moreover, reduced IL-12 production may be related to CD46-downregulation as well as effected by complement and phagocytic receptors on monocytes [41]. As IL-12 is critical for the development of cell mediated immunity and a potent inducer of IFN-γ from T and NK cells, the development of SSPE can pursue in these patients.

On the other hand, a relatively strong immune response generated in the CNS is evident by the anti-MeV antibodies and oligoclonal IgG bands in the CSF and in the serum of majority of SSPE patients. Probably this antibody response is induced by persistent virus in the CNS and with the leakage of the antibodies, some viral antigens can also reach the periphery and cause the subtle changes detected in this study, but not effective in eliminating virus or controlling replication in the CNS. Demonstration of persistent viral RNA in lymphoid cells would have contributed to explain dysfunctional immune response [40].

The study has certain limitations mainly due to the nature of this rare disease. Findings presented here were derived from data obtained throughout many years, which has been a caveat for prospective planning of the experiments in the study. Additionally, the comparisons with two different diseased-control groups have been somehow problematic, as the diseases of the controls were heterogeneous and the donors have been difficult to assign to the respective groups. The two control groups have been composed of children with known diseases with or without inflammatory features.

## Conclusions

In SSPE patients, T cells produced lower levels of IL-10 and IFN-γ, but were inducible to produce IL-2, consistent with an altered immune response of T cells, not competent enough to eliminate the virus in SSPE. Monocytic cells of the patients revealed reduced IL-12 production and CD46 surface expression implicating the effect of CD46 binding in SSPE similar to some MeV strains. Reduced production of IL-10 in combination with reduced IFN-γ points at inefficiency of effector functions of T cells. These observations in SSPE pointed at an attenuated inflammatory pattern at a chronic phase of the disease.

## Supporting information

**S1 Fig. Detection of SLAM on CD19+ B lymphocytes.** SLAM expression on CD19+ cells in a subacute sclerosing panencephalitis patient (SSPE) and in controls with inflammatory diseases (ICON) or non-inflammatory diseases (NICON) are shown.
(TIF)

**S2 Fig. Proliferative responses of T cells to CD3 and CD28 stimulation.** Spontaneous and CD3+CD28 induced proliferative responses of T cells in subacute sclerosing panencephalitis (SSPE) patients and in controls with inflammatory diseases (ICON) and non-inflammatory diseases (NICON) are shown. Cpm: Counts per minute, SI: Stimulation index (SI = Induced cpm / Spontaneous cpm).
(TIF)

**S3 Fig. Detection of CD46 on monocytes.** CD14+CD46+ monocytes in a subacute sclerosing panencephalitis patient (SSPE), controls with inflammatory diseases (ICON) and with non-

inflammatory diseases (NICON) are shown.
(TIF)

**S4 Fig. PD-1 on CD8$^+$ T cells.** PD-1 on CD8$^+$ T cells in a subacute sclerosing panencephalitis patient (SSPE), controls with inflammatory diseases (ICON) and with non-inflammatory diseases (NICON) are shown. Horizontal lines depict median values.
(TIF)

**S1 Data.**
(PDF)

## Acknowledgments

We are grateful to patients and their parents, and to Dr. Erdem Tüzün for critically reading the manuscript.

## Author Contributions

**Conceptualization:** Sibel P. Yentür, Veysi Demirbilek, Candan Gurses, Semih Ayta, Zuhal Yapici, Güher Saruhan-Direskeneli.

**Data curation:** Sibel P. Yentür, Candan Gurses, Safa Baris, Umit Kuru, Zuhal Yapici, Serap Uysal, Gulden Celik Yilmaz, Emel Onal, Ozlem Cokar, Güher Saruhan-Direskeneli.

**Formal analysis:** Semih Ayta, Zuhal Yapici, Güher Saruhan-Direskeneli.

**Investigation:** Sibel P. Yentür, Veysi Demirbilek, Candan Gurses, Semih Ayta, Suzan Adin-Cinar, Güher Saruhan-Direskeneli.

**Methodology:** Safa Baris, Suzan Adin-Cinar, Gulden Celik Yilmaz, Güher Saruhan-Direskeneli.

**Project administration:** Güher Saruhan-Direskeneli.

**Supervision:** Güher Saruhan-Direskeneli.

**Writing – original draft:** Sibel P. Yentür.

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
