## [Decision Letter · Decision Letter 0]

15 Sep 2020

PONE-D-20-23751

Immune alterations in subacute sclerosing panencephalitis reflect an incompetent response to eliminate the measles virus

PLOS ONE

Dear Dr. Saruhan-Direskeneli,

Thank you for submitting your manuscript to PLOS ONE. After careful consideration, we feel that it has merit but does not fully meet PLOS ONE’s publication criteria as it currently stands. Therefore, we invite you to submit a revised version of the manuscript that addresses the points raised during the review process.

As academic editor I received the information from the Journal that methodological and statistical descriptions might not completely meet the standard of the Journal, see below. Please address this point in separately in the revised version.

**Information for the Academic Editor **(as provided by the journal staff):

Note from Associate Editor Susan Hepp (shepp@plos.org): PLOS ONE requires experimental and statistical methods to be described in enough detail to allow suitably skilled investigators to fully replicate and evaluate a study. After internal review, we were concerned that the experimental and statistical methods may not detailed enough to meet these criteria. Please evaluate whether the reporting of experimental and statistical methods meets our submission requirements, which can be found here: (1) https://journals.plos.org/plosone/s/submission-guidelines#loc-materials-and-methods and (2) https://journals.plos.org/plosone/s/submission-guidelines#loc-statistical-reporting.<o:p></o:p>

We look forward to receiving your revised manuscript.

Kind regards,

Edgar Meinl, M.D.

Academic Editor

PLOS ONE

Journal Requirements:

2. Please provide additional details regarding participant consent. In the ethics statement in the Methods and online submission information, please ensure that you have specified:

i) what type of consent you obtained (for instance, written or verbal, and if verbal, how it was documented and witnessed), and

ii) whether informed consent was also obtained from adults in the study.

In addition, we note that you refer to patients as "boys" and "girls" however some patients are not minors. Please revise your manuscript to refer to these patients as "male" and "female".

Also, please clarify whether this study is prospective or retrospective. If this study was retrospective, please include the date(s) on which you accessed the databases or records to obtain the data used in your study. If the study was prospective, please describe the recruitment methods used.

Finally, PLOS ONE requires experimental methods to be described in enough detail to allow suitably skilled investigators to fully replicate and evaluate your study. See https://journals.plos.org/plosone/s/submission-guidelines#loc-materials-and-methods for more information.

To comply with PLOS ONE submission guidelines, in your Methods section, please provide a more detailed description of your methodology, specifically:

a) provide more detailed criteria for the diagnosis of SSPE

b) provide the source, name, and catalog numbers of the ELISA kits for detecting measles

c) provide the source, catalog numbers, and dilutions for all primary/secondary/isotype control antibodies used in the study.

'Funding

This study is supported by Istanbul University Research Fund.'

'The author(s) received no specific funding for this work.'

Reviewers' comments:

Reviewer's Responses to Questions

**Comments to the Author**

1. Is the manuscript technically sound, and do the data support the conclusions?

Reviewer #1: Yes

Reviewer #2: Yes

2. Has the statistical analysis been performed appropriately and rigorously? 

Reviewer #1: Yes

Reviewer #2: Yes

3. Have the authors made all data underlying the findings in their manuscript fully available?

Reviewer #1: Yes

Reviewer #2: Yes

4. Is the manuscript presented in an intelligible fashion and written in standard English?

Reviewer #1: Yes

Reviewer #2: Yes

5. Review Comments to the Author

Reviewer #1: The authors investigated the immune response in patients with SSPE in comparison to two well age-matched control groups consisting of inflammatory (ICON) and non-inflammatory (NICON) diseases. Taking into account that SSPE is a rare disease the number of patients is very high.

The authors used established methods (FACS, ELISA, Elispot assays, 3H thymidine incorporation). PBMCs of patients and controls were investigated ex vivo or after different stimulations in vitro – unspecific (antibodies, SAC) or MeV specific (viral peptides).

The results demonstrate, that PBMC of SSPE patients produced lower levels of IL-10 and IFN-g after stimulation with anti-CD3 and anti CD28, but were inducible to produce IL-2. After SAC stimulation PBMC of SSPE patients showed reduced IL-12p70 production and CD 14+ monocytes demonstrate lower CD46 surface expression. In Elispot assays spontaneous IFN-g production and antigen stimulated IFN-g production was elevated in SSPE patients and NICON compared with ICON.

The authors concluded, that T cells of SSPE patients demonstrate an altered immune response that is not sufficient to eliminate the virus. In monocytes reduced IL-12 production and CD46 surface expression implicate the effect of CD46 binding in SSPE similar to MeV infection.

Major points:

1. The authors report that in a recent epidemiological study of Istanbul a girl dominance was found. In their SSPE cohort there is a dominance of boys. Is this just by chance? From which part of the country or from which country were the patients recruited?

2. Different clinical stages of SSPS are known. For reference see e.g. Jabbour J, et al., SSPE-clinical staging, course, and frequency. Arch Neurol. 1975;32(7):493–494. 24 or Gutierrez J, et al., Dev Med Child Neurol. 2010 Oct;52(10):901-7. It would be interesting to know in which clinical state the patients had been at the time of blood sampling.

3. How do the authors interpret the reduction of T cells in SSPE samples?

4. The authors report “Monocyte stimulation with SAC”. In the legend of Fig. 4 they report “… SAC stimulated PBMC…”. If they don´t select monocytes before stimulation, the headline of this section and the legend of figure 4 should be modified. In addition, the authors should mention that SAC does not stimulate only monocytes in their cell culture.

5. At the beginning of the discussion the authors claim “… no evidence was found for immunosuppressive mechanisms as a determining factor in SSPE development.” The authors may explain more detailed the reasons for this statement or omit it.

6. The authors report a reduced production of IL-10 in the present study, but mention a production of IL-10 as in controls in their previous study. Do the authors have an explanation for this?

7. IL-10 is a cytokine with strong immunosuppressive properties. However, there are also publications demonstrating immunostimulation by IL-10, e.g. Il-10 enhance the capacity of resting CD4+ lymphocytes to produce cytokines. The authors may discuss this aspect as well and not solely the immunosuppressive properties of IL-10.

8. In the section “Antigen-specific T cell stimulation” the authors report “Interestingly, Ifn-g responses to all peptides and peptide pool were reduced in ICON group compared to SSPE (…) and NICON groups (…) …”. The authors may comment on this finding in the discussion, especially as they show an IFN-g response in PBMCs of ICON after unspecific stimulation (Fig. 2).

9. In the discussion the authors report “…SLAM expression was relatively higher in all cell subgroups both in SSPE and other inflammatory diseases compared with donors without inflammation…”. Fig. 3 demonstrate significant differences only for B-cells. This statement has to be modified.

10. The authors report that “… the donors have been difficult to assign to the respective groups.” Please explain the reasons.

Minor points:

1. Determination of CD46 and SLAM should be illustrated with an example showing a FACS analysis of a patient and two controls, e.g. as supplementary figures.

2. In the section “Antigen-specific T cell stimulation” there is a list of p-values. It is unclear to which peptide stimulation a given p-value belongs. The authors should present peptide stimulation and corresponding p-value in a supplementary table or omit the p-values in the text, as they are given in Fig. 6.

3. The citation format of two references in the discussion does not comply with the journal style.

Reviewer #2: In this study, authors aimed to test the cytokine profile of lymphocytes and monocytes obtained from the blood of SSPE patients to gain more insight into the immunopathogenesis of disease. SSPE is a latent brain infection that occurs many years after measles infection and it is a fatal disease. It is an important health problem in some of the developing countries. It is relatively frequent in Turkey, and a high number of SSPE patients were tested in this study. As SSPE is very rare, this study has a potential to contribute to the related literature. However, some points need to be cleared:

Minor revisions:

- Exclusion criteria for all groups should be mentioned.

- How would the authors interpret the finding that IL10 secretion is lower compared to controls? Is this a result of the latent brain infection, or is this more like a risk factor for the disease as a result of genetic factors etc? A comment on that in the discussion would be useful.

- What is the possible functional outcome of the finding that IL12 is decreased in monocytes? A comment on that in the discussion would be useful.

- The authors stated that “In SSPE patients, the lower frequencies ofCD46+ monocytes compared with NICON group in this study also implicated a related interaction of the virus with these cells.” However, SSPE is known as a latent brain infection and active involvement of monocytes in the periphery would be a surprising finding which may refer to ongoing MV activity in peripheral organs. Can the authors make this point clearer in discussion?

- In the conclusion, authors state that their findings pointed at an attenuated inflammatory pattern at a chronic phase of SSPE. Considering that SSPE is a brain-restricted latent infection, what may be the reason behind the attenuated inflammatory pattern seen in the peripheral blood cells? Discussion of this point would be useful.

6. PLOS authors have the option to publish the peer review history of their article (what does this mean?). If published, this will include your full peer review and any attached files.

Reviewer #1: No

Reviewer #2: **Yes: **Atay Vural

---

## [Author Response · Author response to Decision Letter 0]

2 Nov 2020

To comply with PLOS ONE submission guidelines, we have added the description of the methodology in more detail and the supplementary information in supplementary figures (1-4) as requested. We also removed the funding information from the manuscript, which should be included in the manuscript as below: 

The study is supported by Istanbul University Research Fund (BAP: ACIP3363-ACIP-22421-T291).

Responses to Reviewers

Reviewer #1: The authors investigated the immune response in patients with SSPE in comparison to two well age-matched control groups consisting of inflammatory (ICON) and non-inflammatory (NICON) diseases. Taking into account that SSPE is a rare disease the number of patients is very high.

The authors used established methods (FACS, ELISA, Elispot assays, 3H thymidine incorporation). PBMCs of patients and controls were investigated ex vivo or after different stimulations in vitro – unspecific (antibodies, SAC) or MeV specific (viral peptides).

The results demonstrate, that PBMC of SSPE patients produced lower levels of IL-10 and IFN-g after stimulation with anti-CD3 and anti CD28, but were inducible to produce IL-2. After SAC stimulation PBMC of SSPE patients showed reduced IL-12p70 production and CD 14+ monocytes demonstrate lower CD46 surface expression. In Elispot assays spontaneous IFN-g production and antigen stimulated IFN-g production was elevated in SSPE patients and NICON compared with ICON.

The authors concluded, that T cells of SSPE patients demonstrate an altered immune response that is not sufficient to eliminate the virus. In monocytes reduced IL-12 production and CD46 surface expression implicate the effect of CD46 binding in SSPE similar to MeV infection.

Major points:

1. The authors report that in a recent epidemiological study of Istanbul a girl dominance was found. In their SSPE cohort there is a dominance of boys. Is this just by chance? From which part of the country or from which country were the patients recruited?

As pointed out by the reviewer, in our epidemiological study, the distribution of sexes was in favor of the women, whereas our cohort was not in agreement with this observation. The reason for this discrepancy may be related to the natures of the studies. The epidemiology was planned for the determination the incidence in Istanbul area in two years, whereas in our study the patients were collected throughout 14 years in the same area. Some information about the referral system of the patients is also added to the material and methods section.

2. Different clinical stages of SSPS are known. For reference see e.g. Jabbour J, et al., SSPE-clinical staging, course, and frequency. Arch Neurol. 1975;32(7):493–494. 24 or Gutierrez J, et al., Dev Med Child Neurol. 2010 Oct;52(10):901-7. It would be interesting to know in which clinical state the patients had been at the time of blood sampling.

We agree with the reviewer. Unfortunately the information related to the clinical stages of the patients was only available in a small number of patients (n: 44). As the patients were referred from several centers, we could not access their files later on. Patients with known stages were not representing the whole cohort, so that we did not include this information. 

3. How do the authors interpret the reduction of T cells in SSPE samples?

As pointed at by the reviewer, a reduction of T cells in SSPE samples was observed in our previous study (reference 19) and in other studies (reference 30) as well. This observation may be related to the lymphopenia caused by MeV during the infection. MeV infected monocytes induce apoptosis in uninfected T cells which also may contribute to the pathogenesis of MeV-induced immunosuppression (reference 31). We have added this issue to the discussion.

4. The authors report “Monocyte stimulation with SAC”. In the legend of Fig. 4 they report “… SAC stimulated PBMC…”. If they don´t select monocytes before stimulation, the headline of this section and the legend of figure 4 should be modified. In addition, the authors should mention that SAC does not stimulate only monocytes in their cell culture. 

As the reviewer highlighted, the assumption that the reduced IL-12 production is from monocytes was based on the fact that SAC would not be able to induce IL-12 by any other cells type in PBMC. The only differentially produced cytokine induced by SAC was IL-12, so that the monocytes have been accounted for this result. But we agree that the finding is based on an indirect finding and is an interpretation of the data. We have changed the text accordingly.

5. At the beginning of the discussion the authors claim “… no evidence was found for immunosuppressive mechanisms as a determining factor in SSPE development.” The authors may explain more detailed the reasons for this statement or omit it. 

This sentence was meant as a general evaluation of the study and emphasized that the disease was not accompanied by an immunosuppression as observed in measles infection. We have changed this section now accordingly.

6. The authors report a reduced production of IL-10 in the present study, but mention a production of IL-10 as in controls in their previous study. Do the authors have an explanation for this?

We agree with the reviewer that the results of two studies are not in accordance for the spontaneous IL-10 response of the patients. The control group in our previous study was consisted of healthy adults whereas the controls in this study are age-matched diseased controls. The variance between cytokine measurements may be related to this parameter.

7. IL-10 is a cytokine with strong immunosuppressive properties. However, there are also publications demonstrating immunostimulation by IL-10, e.g. Il-10 enhance the capacity of resting CD4+ lymphocytes to produce cytokines. The authors may discuss this aspect as well and not solely the immunosuppressive properties of IL-10.

We agree with the reviewer that IL-10 has pleiotropic roles. However, in infectious diseases, apparently it mainly acts as regulator in a context dependent manner. As the production of IL-10 was reduced in our patient group, we have interpreted these finding as a dysregulation of regulatory response and added a comment to the discussion.

8. In the section “Antigen-specific T cell stimulation” the authors report “Interestingly, Ifn-g responses to all peptides and peptide pool were reduced in ICON group compared to SSPE (…) and NICON groups (…) …”. The authors may comment on this finding in the discussion, especially as they show an IFN-g response in PBMCs of ICON after unspecific stimulation (Fig. 2).

We agree with the reviewer about this comment on the unexpected findings. We consider that the IFN-g response to CD3+CD28 in ICON group is produced by a broad range of T cells as the stimulation is not specific in this case. On the other hand, the peptide specific IFN� production in ICON group may be compromised particularly for the specific responses.

9. In the discussion the authors report “…SLAM expression was relatively higher in all cell subgroups both in SSPE and other inflammatory diseases compared with donors without inflammation…”. Fig. 3 demonstrate significant differences only for B-cells. This statement has to be modified.

The differences of SLAM we observed on CD4, CD8 and CD19 + cells were not significantly different and the only increase was detected on B cells of ICON group. That is why we did not mention the activation state of PMBC in the results section and in the discussion as a comment only. Now we have omitted this statement from the discussion, as it was not based on significant differences.

10. The authors report that “… the donors have been difficult to assign to the respective groups.” Please explain the reasons.

As some of the diseases we included in the controls are not easily assignable to inflammatory or no-inflammatory subgroups. Due to the subtle inflammatory changes in these disease states, we were not sure to include these patients in non-inflammatory cases (X-linked adenoleucodystrophy, nesioblastosis). We have added the some of the diagnosis in these groups.

Minor points:

1. Determination of CD46 and SLAM should be illustrated with an example showing a FACS analysis of a patient and two controls, e.g. as supplementary figures.

FACS analysis of SLAM and CD46 staining is now added as supplementary figures as requested (S1 Fig and S3 Fig).

2. In the section “Antigen-specific T cell stimulation” there is a list of p-values. It is unclear to which peptide stimulation a given p-value belongs. The authors should present peptide stimulation and corresponding p-value in a supplementary table or omit the p-values in the text, as they are given in Fig. 6.

We have omitted the p values from the text.

3. The citation format of two references in the discussion does not comply with the journal style.

We have corrected them.

 

Reviewer #2: In this study, authors aimed to test the cytokine profile of lymphocytes and monocytes obtained from the blood of SSPE patients to gain more insight into the immunopathogenesis of disease. SSPE is a latent brain infection that occurs many years after measles infection and it is a fatal disease. It is an important health problem in some of the developing countries. It is relatively frequent in Turkey, and a high number of SSPE patients were tested in this study. As SSPE is very rare, this study has a potential to contribute to the related literature. However, some points need to be cleared:

Minor revisions:

1. Exclusion criteria for all groups should be mentioned. 

Based on the diagnosis of the patients, we did not exclude any patients with SSPE in the study group. Only patients whose parents did not agree to consent were excluded. All other patients who were not diagnosed with SSPE were included in the control groups according to the disease with or without apparent inflammation. We added this information to the manuscript.

2. How would the authors interpret the finding that IL10 secretion is lower compared to controls? Is this a result of the latent brain infection, or is this more like a risk factor for the disease as a result of genetic factors etc? A comment on that in the discussion would be useful.

We have added a comment to the discussion related to lower IL-10 production in SSPE. We considered the reduced production of IL-10 in combination with reduced IFN-� production as an inefficiency of effector functions of T cells. An exhausted state which we could not demonstrate would be the good explanation for that.

3. What is the possible functional outcome of the finding that IL12 is decreased in monocytes? A comment on that in the discussion would be useful.

As IL-12 is a key inducer of Th1 responses, the decreased activity of Th1 type cells reflected by the lower IFN-� production could be the consequence of this finding. We have added a comment to the discussion.

4. The authors stated that “In SSPE patients, the lower frequencies ofCD46+ monocytes compared with NICON group in this study also implicated a related interaction of the virus with these cells.” However, SSPE is known as a latent brain infection and active involvement of monocytes in the periphery would be a surprising finding which may refer to ongoing MV activity in peripheral organs. Can the authors make this point clearer in discussion?

Although this explanation is not evidence based, we have added some speculative considerations to the discussion about these data. Unfortunately we do not have a good explanation for the development in SSPE in these terms.

5. In the conclusion, authors state that their findings pointed at an attenuated inflammatory pattern at a chronic phase of SSPE. Considering that SSPE is a brain-restricted latent infection, what may be the reason behind the attenuated inflammatory pattern seen in the peripheral blood cells? Discussion of this point would be useful.

With the given data, SSPE is considered as a result of reactivation of a latent infection after many years. The RNA from the measles virus has been isolated in the brain, eyes, and spinal cord in patients with SSPE and persistence of MeV was confirmed in PBMC even after years. An immune response is generated against the virus and a strong antiviral immune response is induced as evidenced by the unusually high levels of antibody in serum and CSF. But it is not effective in eliminating virus or controlling replication in the CNS. During persistence, viral RNA in PBMCs and lymphoid tissue is detected in B, T lymphocytes and monocytes. Persistent RNA in lymphoid cells may contribute to immune response dysfunction by altering the ability of cells to proliferate in response to immune signaling (reference 39).

---

## [Decision Letter · Decision Letter 1]

23 Nov 2020

PONE-D-20-23751R1

Immune alterations in subacute sclerosing panencephalitis reflect an incompetent response to eliminate the measles virus

PLOS ONE

Dear Dr. Saruhan-Direskeneli,

Thank you for submitting your manuscript to PLOS ONE. After careful consideration, we feel that it has merit but does not fully meet PLOS ONE’s publication criteria as it currently stands. Therefore, we invite you to submit a revised version of the manuscript that addresses the points raised during the review process.

We look forward to receiving your revised manuscript.

Kind regards,

Edgar Meinl, M.D.

Academic Editor

PLOS ONE

Reviewers' comments:

Reviewer's Responses to Questions

**Comments to the Author**

1. If the authors have adequately addressed your comments raised in a previous round of review and you feel that this manuscript is now acceptable for publication, you may indicate that here to bypass the “Comments to the Author” section, enter your conflict of interest statement in the “Confidential to Editor” section, and submit your "Accept" recommendation.

Reviewer #1: All comments have been addressed

Reviewer #2: (No Response)

2. Is the manuscript technically sound, and do the data support the conclusions?

Reviewer #1: Yes

Reviewer #2: Partly

3. Has the statistical analysis been performed appropriately and rigorously? 

Reviewer #1: Yes

Reviewer #2: Yes

4. Have the authors made all data underlying the findings in their manuscript fully available?

Reviewer #1: Yes

Reviewer #2: Yes

5. Is the manuscript presented in an intelligible fashion and written in standard English?

Reviewer #1: Yes

Reviewer #2: Yes

6. Review Comments to the Author

Reviewer #1: (No Response)

Reviewer #2: The authors improved the paper in general by providing some corrections and further explanations. However, some points would still be better to be improved:

Point #1: If there were no exclusion criteria, were there any patients or controls who have a concomitant acute/chronic infection (other than SSPE), or autoimmune disorder at the time of blood sampling?

Points #2 and #3: The authors provided appropriate answers.

Point #4: The lower frequencies ofCD46+ monocytes compared with NICON group in the current study is an interesting finding and as the authors state “may implicate a related interaction of the virus with these cells.” Can the authors provide data regarding the presence or absence of MeV RNA in monocytes from SSPE patients?

Point #5:

In response to the point #5, the authors commented that “With the given data, SSPE is considered as a result of reactivation of a latent infection after many years. The RNA from the measles virus has been isolated in the brain, eyes, and spinal cord in patients with SSPE and persistence of MeV was confirmed in PBMC even after years…. During persistence, viral RNA in PBMCs and lymphoid tissue is detected in B, T lymphocytes and monocytes. Persistent RNA in lymphoid cells may contribute to immune response dysfunction by altering the ability of cells to proliferate in response to immune signaling…”

Reference #39 given as the source for this information is a review article and from its references it is seen that persistence of MeV was shown in Macaque Monkeys, HIV-infected children, or shortly after acute infection, but not after many years, or in children with SSPE. However, this is an important and interesting point. Could the authors provide more specific literature, or better, could they provide data regarding the presence or absence of MeV RNA in lymphocytes and also in monocytes, in SSPE children vs controls?

In addition, the authors stated that “An immune response is generated against the virus and a strong antiviral immune response is induced as evidenced by the unusually high levels of antibody in serum and CSF. But it is not effective in eliminating virus or controlling replication in the CNS.”

This is again an interesting and important point. According to the authors’ hypothesis, persistence of MeV and slow replication of the virus in the peripheral lymphoid organs and blood can be a source for the latent brain infection. And generation of a higher titer of anti-measles antibodies would show the presence of the viral persistence, but would be ineffective to eliminate the virus. Can the authors provide the specific literature showing the higher titer of anti-measles IgG in SSPE patients compared to controls? Or better, could they provide the data as an addition to the current study?

7. PLOS authors have the option to publish the peer review history of their article (what does this mean?). If published, this will include your full peer review and any attached files.

Reviewer #1: No

Reviewer #2: No

---

## [Author Response · Author response to Decision Letter 1]

16 Dec 2020

PONE-D-20-23751R2

Responses to the reviewer

Point #1: If there were no exclusion criteria, were there any patients or controls who have a concomitant acute/chronic infection (other than SSPE), or autoimmune disorder at the time of blood sampling?

There were no SSPE patients with other disorders according to the assessment of the following clinician. We have added a remark about the problem of data acquisition from the patients in the methods section.

Point #4: The lower frequencies of CD46+ monocytes compared with NICON group in the current study is an interesting finding and as the authors state “may implicate a related interaction of the virus with these cells.” Can the authors provide data regarding the presence or absence of MeV RNA in monocytes from SSPE patients?

In this study, although we intended to identify the virus in the patient material in the beginning, we were not able to assess the MeV RNA in the samples of SSPE patients. However, the presence of viral RNA has been documented in monocytes during acute measles infection (Reference 39). MeV infection has been also shown to suppress IL-12 secretion from monocytes (Reference 15). Assuming that the virus used CD46 to enter the monocytes, we claimed that the decreased IL-12 production and CD46 expression could fit well with this picture. Unfortunately, there is no evidence of the viral persistence in the peripheral blood cells, including monocytes, shown formally in SSPE. We have changed the section accordingly.

Point #5:

In response to the point #5, the authors commented that “With the given data, SSPE is considered as a result of reactivation of a latent infection after many years. The RNA from the measles virus has been isolated in the brain, eyes, and spinal cord in patients with SSPE and persistence of MeV was confirmed in PBMC even after years…. During persistence, viral RNA in PBMCs and lymphoid tissue is detected in B, T lymphocytes and monocytes. Persistent RNA in lymphoid cells may contribute to immune response dysfunction by altering the ability of cells to proliferate in response to immune signaling…”

Reference #39 given as the source for this information is a review article and from its references it is seen that persistence of MeV was shown in Macaque Monkeys, HIV-infected children, or shortly after acute infection, but not after many years, or in children with SSPE. However, this is an important and interesting point. Could the authors provide more specific literature, or better, could they provide data regarding the presence or absence of MeV RNA in lymphocytes and also in monocytes, in SSPE children vs controls?

The hypothesis that “the persistence of MeV RNA may contribute to the late development of the slowly progressive disease subacute sclerosing panencephalitis in children infected at a young age” (reference 22) has been proposed by the authors studying measles infection. We also found this approach somehow suitable to our findings. As pointed at by the reviewer, there are no data on the presence of the viral RNA in any peripheral cell or tissue in SSPE. However, based on the proposed similarities between the monkey model and the humans in terms of measles (Measles virus infection in rhesus macaques: Altered immune responses and comparison of the virulence of six different virus strains. J Infect Dis. 1999;180:950–958), we assumed that this explanation can also be applied to our data. Now we have changed the section accordingly. Unfortunately, we do not have any data about the presence of MeV RNA in any cell group of SSPE patients in this study group. 

In addition, the authors stated that “An immune response is generated against the virus and a strong antiviral immune response is induced as evidenced by the unusually high levels of antibody in serum and CSF. But it is not effective in eliminating virus or controlling replication in the CNS.”

This is again an interesting and important point. According to the authors’ hypothesis, persistence of MeV and slow replication of the virus in the peripheral lymphoid organs and blood can be a source for the latent brain infection. And generation of a higher titer of anti-measles antibodies would show the presence of the viral persistence, but would be ineffective to eliminate the virus. Can the authors provide the specific literature showing the higher titer of anti-measles IgG in SSPE patients compared to controls? Or better, could they provide the data as an addition to the current study?

We thank the reviewer for raising this point. Unfortunately, we were not precise in our wording: what we know about the antibody response in SSPE is that anti-MeV antibodies are present in the CSF of patients and this is utilized even as a diagnostic finding for SSPE (Reference 16 and 17). The clinical diagnosis of SSPE is confirmed by the finding of increased levels of anti-measles virus anti- body in serum and CSF with an elevated CSF-to- serum ratio in antibody level. Actually, most of the time the antibodies are also detected in the plasma. In our own experience, we have screened the oligoclonal IgG bands in 108 of these study patients. The positivity of IgG bands in the CSF was 100% with a few atypical patterns and all OCB positive patients had also anti-MeV antibodies in the CSF. In 82.4 % of the patients, serum samples also contained accompanying and presumably leaked IgG bands (pattern 3 in 79.6%, pattern 4 in 2.8%). Although these antibodies were not tested for their specificity for MeV, there are data supporting that these antibodies are MeV specific. Our interpretation of the peripheral anti-viral antibody was based on the oligoclonal bands in the patients. The findings only allow to explain the immune response in periphery induced by leaked antigens and antibodies from the CNS. We have also changed the section in the discussion accordingly.

---

## [Decision Letter · Decision Letter 2]

22 Dec 2020

Immune alterations in subacute sclerosing panencephalitis reflect an incompetent response to eliminate the measles virus

PONE-D-20-23751R2

Dear Dr. Saruhan-Direskeneli,

We’re pleased to inform you that your manuscript has been judged scientifically suitable for publication and will be formally accepted for publication once it meets all outstanding technical requirements.

Kind regards,

Edgar Meinl, M.D.

Academic Editor

PLOS ONE

Additional Editor Comments (optional):

Reviewers' comments:

Reviewer's Responses to Questions

**Comments to the Author**

1. If the authors have adequately addressed your comments raised in a previous round of review and you feel that this manuscript is now acceptable for publication, you may indicate that here to bypass the “Comments to the Author” section, enter your conflict of interest statement in the “Confidential to Editor” section, and submit your "Accept" recommendation.

Reviewer #2: All comments have been addressed

2. Is the manuscript technically sound, and do the data support the conclusions?

Reviewer #2: Yes

3. Has the statistical analysis been performed appropriately and rigorously? 

Reviewer #2: Yes

4. Have the authors made all data underlying the findings in their manuscript fully available?

Reviewer #2: Yes

5. Is the manuscript presented in an intelligible fashion and written in standard English?

Reviewer #2: Yes

6. Review Comments to the Author

Reviewer #2: (No Response)

7. PLOS authors have the option to publish the peer review history of their article (what does this mean?). If published, this will include your full peer review and any attached files.

Reviewer #2: No

---

## [Editor Report · Acceptance letter]

29 Dec 2020

PONE-D-20-23751R2 

Immune alterations in subacute sclerosing panencephalitis reflect an incompetent response to eliminate the measles virus 

Dear Dr. Saruhan-Direskeneli:

I'm pleased to inform you that your manuscript has been deemed suitable for publication in PLOS ONE. Congratulations! Your manuscript is now with our production department. 

Kind regards, 

on behalf of

Prof Edgar Meinl 

Academic Editor

PLOS ONE